# Metalloproteinases and Hypertrophic Cardiomyopathy: A Systematic Review

**DOI:** 10.3390/biom13040665

**Published:** 2023-04-11

**Authors:** Giuseppe Filiberto Serraino, Federica Jiritano, Davide Costa, Nicola Ielapi, Desirèe Napolitano, Pasquale Mastroroberto, Umberto Marcello Bracale, Michele Andreucci, Raffaele Serra

**Affiliations:** 1Department of Experimental and Clinical Medicine, “Magna Graecia” University of Catanzaro, 88100 Catanzaro, Italy; 2Interuniversity Center of Phlebolymphology (CIFL), International Research and Educational Program in Clinical and Experimental Biotechnology, University Magna Graecia of Catanzaro, 88100 Catanzaro, Italy; 3Department of Public Health and Infectious Disease, “Sapienza” University of Rome, 00185 Roma, Italy; 4Ph.D. Student “Digital Medicine” Ph.D. Programm-Magna Graecia, “Magna Graecia” University of Catanzaro, 88100 Catanzaro, Italy; 5Department of Public Health, Vascular Surgery Unit, University of Naples “Federico II”, 80126 Naples, Italy; 6Department of Health Sciences, Nephrology Unit, University of Catanzaro, 88100 Catanzaro, Italy; 7Department of Medical and Surgical Sciences, “Magna Graecia” University of Catanzaro, 88100 Catanzaro, Italy

**Keywords:** matrix metalloproteinases, hypertrophic cardiomyopathy, collagen, extracellular matrix, biomarkers

## Abstract

Hypertrophic cardiomyopathy (HCM) is a genetic condition determined by an altered collagen turnover of the extracellular matrix. Matrix metalloproteinases (MMPs) and their inhibitors (TIMPs) are abnormally released in patients with HCM. The purpose of this systematic review was to thoroughly summarize and discuss the existing knowledge of MMPs profile in patients with HCM. All studies meeting the inclusion criteria (detailed data regarding MMPs in patients with HCM) were selected, after screening the literature from July 1975 to November 2022. Sixteen trials that enrolled a total of 892 participants were included. MMPs–particularly MMP2—levels were found higher in HCM patients compared to healthy subjects. MMPs were used as biomarkers after surgical and percutaneous treatments. Understanding the molecular processes that control the cardiac ECM’s collagen turnover allows for a non-invasive evaluation of HCM patients through the monitoring of MMPs and TIMPs.

## 1. Introduction

Hypertrophic cardiomyopathy (HCM) is a hereditary condition. Historically, it was known as idiopathic hypertrophic subaortic stenosis [1]. Clinical signs and symptoms of HCM can range from asymptomatic disorders to progressive cardiac failure [2]. HCM is also a substantial factor in sudden cardiac mortality in young people, even well-trained athletes, equally affecting both sexes and regardless of race [3,4,5]. A common finding in HCM patients is left ventricular outflow obstruction brought on by asymmetric septal hypertrophy. The interventricular septum is where the hypertrophy is most common, though it can happen in any area of the left ventricle. The left ventricular outflow tract is frequently obstructed as a result. Septal hypertrophy is caused by genetic flaws in a number of different genes [1,3,4,5].

Histology may show a severe disorder of the architecture, cellular disarray, and disorganization of the muscle fibers. Fibrosis could be visible. The diameter of the coronary arteries may be reduced and they may be intramural [6]. The extracellular matrix (ECM) is one potential source of self-sustaining hypertrophic stimulation in HCM hearts. Myocardial fibrosis is common in established HCM, regardless of severity. Evidence of aberrant ECM turnover has also been observed in preclinical HCM-mutant subjects [7], indicating that ECM alterations manifest early and may be important in disease development.

The equilibrium between protein synthesis and degradation preserves the ECM’s structural integrity. Fibroblasts release myocardial matrix metalloproteinases (MMPs). The MMPs are a group of protease enzymes that, despite having various substrates, structurally have a lot in common. Three histidine residues are bound to catalytic zinc in the zinc-dependent and highly conserved active region. They work perfectly at a neutral pH. MMPs’ main biological role is the breakdown of membrane receptors, cytokines, growth factors, and ECM proteins and glycoproteins. The MMPs participate in a wide range of biological processes, including tissue regeneration and remodelling, cellular differentiation, embryogenesis, morphogenesis, cell mobility, angiogenesis, cell proliferation, and migration, wound healing, apoptosis, and major reproductive events like ovulation and endometrial growth. MMPs and their inhibitors (tissue inhibitor of metalloproteinase—TIMPs) are co-expressed in the cardiac muscle [8]. Consequently, there is an endogenous inhibitory mechanism in the heart, as well as other tissues, which suggests that MMP activity is regulated by their inhibitors at both the gene and protein levels. Additionally, MMPs and their inhibitors can be individually controlled or together to preserve the interstitial tissue’s architecture. As a result, the amount of real proteolysis can rely on how MMPs and TIMPs are balanced. It is still unclear how MMPs’ overall activity relates to the remodelling of the heart chambers. Additionally, there is ambiguity about the clinical application of MMPs as cardiac remodelling biomarkers. We performed the present systematic review to address this issue and provide an overview of the existing literature on the role of MMPs in the pathophysiology of HCM patients.

## 2. Material and Methods

### 2.1. Protocol

The review was performed in accordance with instructions given by the Cochrane Handbook for Systematic Reviews of Interventions [9]. The systematic review was performed according to PRISMA (Preferred Reporting Items for Systematic Reviews and Meta-Analyses) guidelines. The analysis was registered on Open Science Framework (https://doi.org/10.17605/OSF.IO/YF7B9, accessed on 29 March 2023). Search methods, data extraction, assessment, and presentation were performed as recommended by the Cochrane Handbook for Systematic Reviews of Interventions (Version 5.1).

### 2.2. Eligibility Criteria

Randomized (RCTs) and non-randomized controlled trials (NRCTs), as well as prospective and retrospective observational cohort studies, irrespective of blinding, language, publication status, and date of publication, were considered eligible for this study. Participants of any age undergoing cardiac surgery with cardiopulmonary bypass were considered. Studies were not included in the analysis if they met one of the following exclusion criteria: (i) the analysis was a review, case report, case series (<10 patients), or a conference abstract; or (ii) the analysis provided incomplete information about study objectives. Pre-clinical studies (non-human studies) were excluded from the main analysis. Inclusion and exclusion criteria for qualitative/quantitative analyses were summarized according to the PICo (population/patient/problem, interest, contest) approach (Appendix A).

### 2.3. Subject of Interest

We assessed trials evaluating the role of metalloproteinases in subjects that had undergone cardiac surgical operations with cardiopulmonary bypass.

### 2.4. Information Sources

Potentially eligible studies were identified after an extensive search of the literature conducted through PubMed and Scopus without date or language restrictions. Key words and MeSH terms pertinent to the exposure of interest were used in relevant combinations: “metalloproteinases”, “metalloproteases”, “TIMP”, “cardiopulmonary bypass”, and “hypertrophic cardiomyopathy”. The literature search was run from July 1975 to November 2022. In addition, we searched trial registries, and reference lists were carefully analysed for pertinent studies. Case reports, opinions, and editorials were excluded.

### 2.5. Study Selection and Data Items

Two reviewers (F.J., G.F.S.) independently identified trials for inclusion. Excluded studies and reasons for exclusion were recorded. Two authors independently screened the search output to identify records of potentially eligible trials examining outcomes, the full texts of which were retrieved and assessed for inclusion. A standardized form was used to extract data from the included studies for the assessment of study quality and evidence synthesis. Extracted information included the following: year of publication; study population, with inclusion and exclusion criteria; sample size; participant characteristics; baseline characteristics; outcomes; and information for assessment of the risk of bias. Data extraction forms were completed by one author and checked by a second author. Similarly, quality assessment was performed by one author and checked by a second.

### 2.6. Risk of Bias in Individual Studies

The methodological quality of randomized trials was assessed using the Cochrane Collaboration’s tools for assessing risk of bias in parallel group and cluster randomized trials [9,10]. The items assessed for parallel group trials were as follows: (i) sequence generation; (ii) allocation concealment; (iii) blinding of outcome assessor; (iv) incomplete outcome data; (v) selective outcome reporting; and (vi) other sources of bias, including funder bias. Risk of bias was graded as unclear, high, or low. We graded sealed opaque envelopes as unclear evidence of allocation concealment. We also considered the absence of a prespecified protocol or trial registration of trial design as unclear evidence of reporting bias. The risk of bias in cluster-randomized trials was assessed as follows: (i) recruitment bias; (ii) baseline imbalance; (iii) loss of clusters, incorrect analysis; and (iv) comparability with individually randomized trials.

For NRCTs, a modified Newcastle–Ottawa quality assessment scale for cross-sectional studies was used to assess the quality of the study for inclusion. The total score for the modified Newcastle–Ottawa scale for cross-sectional studies is nine (9) stars as a maximum for the overall scale, with a minimum of zero. A study was considered high quality if it achieved 7 out of 9 and medium quality if it achieved 5 out of 9 (Appendix A). Overall quality was independently determined by each reviewer, with discrepancies solved by consensus.

## 3. Results

A total of 175 abstracts were retrieved from the searches (Appendix A). There were 97 articles screened and 77 excluded. A total of 20 relevant publications were retrieved for further assessment. Furthermore, 16 trials that enrolled a total of 892 participants met the inclusion criteria and were included in the systematic review [10,11,12,13,14,15,16,17,18,19,20,21,22,23,24,25]. Two review authors (F.J., G.F.S.) agreed on the selection of included studies. Key characteristics of individual studies are described in Table 1.

### 3.1. Included Studies

Of the sixteen studies included, two were retrospective NRCTs (n = 95) [17,21], whereas the remaining fourteen were prospective observational NRCTs [10,11,12,13,14,15,16,18,19,20,22,23,24,25]. All studies included patients with HCM, but some better defined their population: two studies enrolled patient undergoing alcohol septal ablation (n = 85) [10,14], one study included patients submitted to percutaneous intramyocardial septal radiofrequency ablation [24], and two enrolled patients submitted to surgical myectomy (n = 107) [23,25].

Seven studies compared HCM patients with a healthy cohort [11,12,13,14,15,20,22].

The present systematic review showed that the MMPs analysed in patients with hypertrophic cardiomyopathy where the following (Table 2):

Interstitial collagenases: MMP-1, MMP-8 (neutrophil collagenase or collagenase-2), MMP-13 (collagenase 3).

Gelatinases: MMP-2 (gelatinase A), MMP-9 (gelatinase B).

Stromelysins: MMP-3 (stromelysin-1).

Tissue inhibitor of matrix metalloproteinase: TIMP-1, TIMP-2, TIMP-3, and TIMP-4.

The gelatinases MMP-2 and MMP-9 were investigated in most of the trials. MMP-2 was the metalloproteinase investigated in thirteen trials [10,11,13,15,16,17,18,19,20,21,23,24,25], whereas twelve trials studied MMP-9 [10,11,13,15,17,18,19,20,21,22,23,25].

Interstitial collagenases were described in five studies: MMP-1 was studied in five trials [10,12,15,19,21], while MMP-8 and MMP-13 were investigated in one prospective observational NRCT [10]. The stromelysin MMP-3 was analysed in three trials [13,19,21].

Thirteen studies analysed TIMP-1 [10,11,12,13,15,16,17,18,19,20,21,23,25], three trials investigated TIMP-2 [13,19,20], one study described TIMP-3 [19], while two papers evaluated TIMP-4 [14,19].

### 3.2. Excluded Studies

Four trials that met our inclusion criteria were excluded after review of the full manuscript (Appendix A) because they were animal research [26,27,28,29].

### 3.3. Main Findings

MMPs levels were found to be higher in six of seven studies comparing HCM patients with healthy subjects, reflecting that collagen turnover is enhanced in HCM patients [11,12,13,14,15,22]. Among all the MMPs, MMP-2 was found to be increased in HCM patients with low ejection fraction (EF) [13,17,18]. Moreover, it was associated with left atrial and left ventricular dimensions [16,17]. MMP-2, in particular, was positively correlated with the myocardial collagen volume fraction [23]. MMP-2 was usually increased after a treatment [10], but it was found to be lower than the baseline at 1-month follow-up [24]. Among TIMPs, TIMP-1 was overexpressed in HCM subjects and was found to be associated with cardiac chambers remodelling [10,11,12,13,16,17]. MMP-3 was observed to be associated with cardiac events such as syncope, ventricular arrythmia and sudden death in HCM patients [19,21].

Five studies analysed MMPs in HCM patients treated with surgical myectomy [23,25], alcohol septal ablation [10,14], and percutaneous intramyocardial radiofrequency septal ablation [24]. In patients submitted for surgical myectomy, plasma MMP-2 levels reflected myocardial fibrosis [23,25]. Patients submitted for alcohol septal ablation showed increased MMP-2, MMP-8, MMP-9, TIMP-1, and TIMP-4 immediately after the procedure [10,14], whereas those submitted for percutaneous radiofrequency intramyocardial septal ablation showed lower MMP-2 levels than the baseline at 1-month follow-up [24].

### 3.4. Risk of Bias and Study Quality

A summary of the risk of biases of the included trials is reported in the Appendix A. All studies were NRCTs. Quality assessment for observational studies showed seven high-quality studies [11,12,13,14,15], one medium-quality study [24], and eight low-quality studies [10,16,17,18,19,21,23,25].

## 4. Discussion

To the best of the authors’ knowledge, this systematic review is the first to offer a thorough explanation of the role of metalloproteinases in HCM patients.

Increased collagen in the myocardial interstitium, unbalanced proportions, and an unorganized arrangement of collagen components are the primary pathological characteristics of HCM. Functionally, it is primarily defined by increased myocardial stiffness, abnormal coronary reserve function, and ventricular systolic and diastolic dysfunction. HCM is a hereditary cardiac myocyte ailment characterized by unexplained ventricular hypertrophy, a nondilated left ventricle, and a normal or enhanced ejection fraction. Cardiac hypertrophy is typically asymmetrical, with the basal interventricular septum next to the aortic valve most frequently affected. It is occasionally also limited to other myocardial areas, such as the left ventricle’s apex, midsection, and posterior wall [30]. The left ventricular wall irregularly thickens in HCM, and the cardiomyocyte hypertrophy and fiber disarray are unevenly distributed throughout the ventricular wall. Interstitial fibrosis leads the cardiomyocytes to become hypertrophied, disorganized, and separated. Furthermore, aberrant intramyocardial coronary arterioles and arteries, which are also present in the non-hypertrophied wall segments, are documented, as well as an elevated collagen content that results in an unstructured extracellular matrix (ECM) [30].

ECM produced by cardiac fibroblasts is primarily made up of type I (85%) and type III (11%) collagen and is crucial for myocardial remodelling [31]. The fibrillar collagens offer a three-dimensional scaffolding that stabilizes the ventricle geometry and keeps the cardiomyocytes aligned, whereas glycoproteins and proteoglycans—non-fibrillar components—contribute to the elasticity and resistance of the tissue [32]. Cardiomyocytes’ lateral cell surfaces are connected by collagen struts and are covered in a collagenous texture. The equilibrium of collagen synthesis and breakdown is influenced by collagenolytic MMPs.

### 4.1. Matrix Metalloproteinases and Their Role in Cardiac Tissue

The MMPs are a family of proteolytic enzymes that break down the ECM. Collagenases, gelatinases, stromelysins, metrilysins, and membrane-type MMPs are additional subgroups of MMPs that can be distinguished based on substrate specificity, sequential similarity, and domain structure

Collagenases cleave fibrillar collagen types I, II, III, IV, and XI into two distinctive pieces, 1/4 C-terminal and 3/4 N-terminal. MMP first unravels triple helical collagen, and then the peptide links are hydrolyzed in this two-step process. While the catalytic domain can also cleave non-collagen substrates, the hemopexin domain is required for cleaving native fibrillar collagen.

Gelatinases are crucial for a variety of physiological processes, including osteogenesis, ECM remodeling, and wound repair. Due to the presence of three repetitive fibronectin type II domains, which bind gelatin, collagen, and laminin, gelatinases degrade gelatin, collagen type IV, V, VIII, X, and XIV, elastin, proteoglycan core proteins, fibronectin, laminin, fibrilin-1, and TNF- and IL-1b precursors. While acting weaker than collagenase, MMP-2 is mainly a gelatinase. Collagen is broken down by MMP-2 in two steps: first, a mild interstitial collagenase-like collagen degradation is induced; next, the fibronectin-like domain is used to promote gelatinolysis. MMP-9 has gelatinase and collagenase activity.

Despite not cleaving interstitial collagen, stromelysines have the same domain structure as collagenases. MMP-3 and MMP-10 are structurally and substrate-specifically linked, whereas MMP-11 is not.

The primary distinction between matrilysins and other MMPs is the absence of the hemopexin region. The amino acid sequence of this MMP group has a distinctive characteristic with a threonine residue close to the Zn^2+^-binding site.

Approximately 30 MMP members are currently known to us. MMP-2 and MMP-9 have attracted the most research interest among the MMPs examined in the heart. Initially, this was due to the technical fact that MMP-2 and MMP-9 are the only MMPs that can be seen by gelatin zymography. In addition, before MMP antibodies were commercially available, these two MMPs were the most straightforward to assess and, as a result, were the most commonly studied. Due to their strong molecular relationship with cardiac remodelling, MMP-2 and MMP-9 are still being studied. As confirmed by the present study, MMP-2 and MMP-9 are the most studied as biomarkers of cardiac remodelling [10,11,13,15,16,17,18,19,20,21,22,23,24,25]. The heart’s cardiomyocytes, fibroblasts, endothelial cells, and inflammatory cells can all make and secrete MMP-2. Because MMP-2 can break down myosin light chain 1 and troponin I, two components of the contractile apparatus, cardiac contractility is decreased. MMP-9, another gelatinase, has similar properties. It is synthesized by myocytes, fibroblasts, and smooth muscle cells. MMP-9 processes both ECM substrates such as denatured collagens, fibronectin, and laminin, as well as non-ECM substrates such as interleukin (IL)-1, IL-6, and dormant transforming growth factor-beta (TGF-). MMP-9 regulates the pathogenesis of many diseases, including cardiac remodelling, by acting on a diverse variety of substrates.

The TIMPs take part in the endogenous system, which post-translationally controls the action of MMPs in a variety of tissues, including the myocardium. The amino-terminal domain of TIMPs, which tightly bind to active enzyme sites to create 1:1 complexes, interacts with active forms of MMPs to inhibit them. Four TIMPs have so far been cloned, and three of them—TIMP2, TIMP3, and TIMP4—are expressed in healthy hearts. TIMP1, however, is expressed at low amounts in healthy hearts, but more abundantly in hearts with diseases. TIMPs have the ability to block various MMPs, while also having individual characteristics. The four known types of TIMPs can bind to active and, in some situations, inactive MMPs with remarkable efficiency. In healthy cardiac tissue, collagen synthesis and degradation are balanced. Any discrepancy between proteinase and proteinase inhibitors is thought to affect the architecture of the myocardial tissue.

### 4.2. Cardiac Fibrosis and Matrix Metalloproteinases

There are a number of hypotheses about the causes of heart fibrosis in HCM, but two of them have the strongest evidence: (1) the higher collagen turnover causes the ECM to switch from collagen I to collagen III; (2) replacement fibrosis is brought on by ischemic myocytic necrosis (Figure 1).

#### 4.2.1. The Collagen Switch Hypothesis

As the matrix metalloproteinase system is implicated in both ventricular hypertrophy and dilation, they may be linked to left ventricular remodelling and the development of left ventricular failure in HCM patients [15]. ECM remodelling is a critical component of disease-related heart remodelling. The link between myocardial cells and blood vessels is disrupted when the ECM network structure is damaged, compromising the heart’s structural stability and functionality. Conversely, increased myocardial stiffness brought on by an excess of ECM structural protein synthesis and accumulation, or fibrosis, prevents ventricular contraction and relaxation, and distorts the structure and function of the heart. Myocardial stiffness, diastolic dysfunction, and fibrosis caused by excessive collagen accumulation have all been associated. Fibrillar collagen types I and III, the main elements of cardiac ECM, are generated as pro-collagens, which are then transformed into mature collagen molecules after procollagen peptidase cleaves their pro-peptide region. Collagen fibrils and collagen strands are created through the assembly and cross-linking of mature collagen molecules. Collagen strands are destroyed and the telopeptides in the amino-terminals or carboxy-terminals of collagen molecules are cleaved during physiological ECM turnover or pathological ECM remodelling. Since the pro-peptides from collagen type I’s carboxy-terminal and type III’s amino-terminal propeptides (PICP, PINP) are released during the biosynthesis of these collagens in a stoichiometric way, they are regarded as biomarkers of collagen synthesis. However, the carboxy-terminal or amino-terminal telopeptide of type I (CITP, NITP) and type III (CIIITP, NIIITP), which are generated when these collagens are degraded, are thought to be indicators of collagen degradation. According to many papers, an enhanced collagen turnover has occurred in HCM patients [11,12,13,15,17,20,22,23]. However, results from earlier research on the function of these biomarkers of myocardial fibrosis are contradictory. Lombardi and colleagues discovered that serum CITP levels were higher in HCM cases, but not in the controls [11]. Conversely, Ho and associates discovered that HCM patients had greater serum PICP levels, but not higher CITP levels [7].

By hydrolysing the peptide bond that follows a Gly residue that is situated three-fourths of the length of the collagen molecule from the amino terminus, MMP-1 starts the digestion of collagens. MMP-2 and -9, along with MMP-1 and -3, totally degrade the resulting three-fourths and one-fourth fragments. MMP-2 and MMP-9 levels have been noticeably greater in HCM patients compared to healthy subjects [11,13,15]. The signal chain that initiates cardiac remodelling is triggered by early MMP-9 activation, whereas MMP-9 inhibition reduces collagen accumulation and left ventricular remodelling [15]. This finding suggests that the ECM in HCM has been activated. Collagen I is simultaneously degrading and collagen III production is increasing in HCM patients. This indicates a change from collagen I to collagen III in the extracellular matrix structure. These changes may lead to alterations in the mechanical properties of the tissue. A compensatory mechanism to the rise in wall stiffness (due to hypertrophy, disarray, and interstitial fibrosis) could be proposed because collagen I is thought to be more rigid than collagen III [33]. Moreover, as the result of a feedback regulation aiming to balance increased MMP activity and collagen I breakdown, TIMPs levels have usually been increased [12,13,14,20,23,25]. Due to the reduced motility of mutant contractile proteins, contractility is decreased at the cellular level in HCM; this impairment may result in the local release of trophic factors (such as tumor necrosis factor, endothelin-1, transforming growth factor 1, and insulin-like growth factor 1), which activate the extracellular matrix, causing interstitial fibrosis and changing the scaffold on which myocytes align, as a compensatory mechanism.

The MMP/TIMP system consequently fails. The heart’s fibroblasts become hyperactive and transdifferentiate into myofibroblasts, which boost collagen production. The rate of collagen breakdown slows down, leading to aberrant collagen accumulation in the myocardium. Myocardial tissue composition adverse changes can be even seen in the early stages of HCM in young people, where MMP-9 levels were increased and correlated to myocardial ECM remodelling [22].

#### 4.2.2. The Ischemic Myocytic Necrosis Hypothesis

On the other hand, a second theory hypothesizes that ischemic myocytic necrosis is the cause of replacement fibrosis. One of the major pathological traits of HCM patients is microvascular remodelling, which includes structural anomalies of intramural coronary arterioles and microvascular rarefaction. Most likely, the abnormal alterations in the microvascular system reflect hypoperfusion under stress, which creates an ischemic substrate that favours more serious fibrosis. Regardless of the underlying mutation, myocardial fibrosis has been seen in a variety of transgenic mouse models of HCM, providing evidence of either a primary independent process or a secondary effect brought on by myocyte hypertrophy, along with impaired capillary density that may be brought on by insufficient vascular growth. Myocardial hypoxia can stimulate fibroblasts, leading to collagen build-up [34]. Patients with HCM have abnormal intramural and subendocardial coronary arteries and arterioles that have been linked to myocardial perfusion anomalies [35], which can result in myocardial ischemia and the classic clinical signs of angina pectoris, lactate production, and scarring [36]. Therefore, the frequently occurring myocardial ischemia and micro-necrosis followed by collagen accumulation could be caused by the reported microvascular dysfunction, which is clinically understood by impaired coronary reserve. As usually occurs after an acute myocardial infarction, MMPs regulates cardiac remodelling progression. Ischemia brought on by microvascular rarefaction raises MMP expression, but it also stimulates collagen synthesis; since there are more MMPs than collagen can be broken down, collagen accumulates [25]. Therefore, given the complexity of myocardial fibrosis development in HCM patients, cardiac fibrosis relief may depend more on inhibiting collagen synthesis than simply improving microvascular function [25].

Furthermore, increased chamber stiffness brought on by abnormal collagen accumulation explains diastolic heart failure symptoms. Several studies correlated MMPs/TIMPs ratios unbalance with adverse cardiac remodelling and cardiac symptoms such as dyspnoea, ventricular arrythmias, and sudden death [15,16,17,18,19,20,21]. Indeed, two studies reported positive correlation between high MMP-3 concentrations and ventricular arrythmias [19,21].

### 4.3. Matrix Metalloproteinases in the Treatment for Hypertrophic Cardiomyopathy

Thus, myocardial fibrosis and collagen deposition may worsen prognosis, be proarrhythmic and cause sudden cardiac death, or have hemodynamic effects that cause diastolic heart failure. There are currently few possibilities for therapy. Treatment options include surgical myectomy, endocardial radiofrequency, or alcohol ablation of septal hypertrophy [2]. Hence, MMPs analysis has expanded throughout different HCM treatments. Bradham and associates reported the altered temporal profile in MMPs plasma levels after alcohol septal ablation [10]. In a small prospective observational study, Qian and coworkers used MMP-2 as biomarkers in patients submitted for percutaneous intramyocardial septal radiofrequency ablation [24]. At one-month follow-up, MMP-2 plasma levels were lower than before the procedure [24]. Similarly, Yang and colleagues studied the effect of medical therapy on HCM patients [23]. They observed that calcium channel blockers may attenuate cardiac fibrosis in patients with HCM [23]. Patients taking diltiazem or verapamil, in fact, had significantly lower levels of products of collagen metabolism [23]. However, not all patients, or only a small percentage of them, responded to these medications [37]. Finding new therapeutic targets is essential for the successful treatment of HCM, and MMPs may be one of them. Liu and colleagues studied the effect of resveratrol—the chemical name of which is 3,5,4′-trihydroxy-trans-stilbene—on patients with cardiac hypertrophy [38]. Resveratrol inhibits cardiac hypertrophy by decreasing blood pressure, but it also affects other variables in addition to the shift in hemodynamic load. Activating the anti-hypertrophic AMP-activated protein kinase (AMPK) signal pathway and inhibiting the hypertrophic Akt signal pathway are two possible examples of these processes. In addition to inhibiting hypertrophy, AMPK can postpone the progression of cardiac enlargement into heart failure. Notably, AMPK can prevent angiotensin II-induced myocardial fibrosis, which inhibits cardiac remodelling. Although resveratrol has numerous pharmacological effects, including antifibrosis, anti-inflammatory, antioxidative, lipid-lowering, and hypoglycaemic effects, and exerts an antifibrosis effect via a variety of growth factors, cytokines, and cell signalling pathways, research on its role in cardiac fibrosis is insufficient and calls for further investigation.

### 4.4. Strengths and Limitations

The present systematic review is the most thorough analysis of MMPs in HCM patients to date. It employed in-depth search strategies across numerous registries and data sources, had access to the full texts of all reported trials, applied up-to-date risk-of-bias assessments, and assessed clinical outcomes. The main limitation of this systematic review’s findings and interpretations is the quality and quantity of available evidence. The review highlighted substantial serious limitations in all of the studies, which were shortcomings of the existing data. The possibility of procedural bias was high, and the way that researchers presented their findings substantially varied. The generalizability of the results of these studies on MMPs in the population of patients with HCM is further constrained by the limited availability of robust randomized data, as well as the limited clinical validation of findings. The manuscript is affected by a significant amount of heterogeneity between studies due to the analysis of different MMPs, various outcome assessment measures, and different approaches to performing these measurements. Meta-analysis was not performed because of the heterogeneity and consequent lack of consistency in stated results.

### 4.5. Clinical Importance

It is crucial to thoroughly comprehend the pathophysiology behind HCM in order to develop therapeutic strategies to avoid unfavourable clinical outcomes. To date, it has become clear that any measures to reduce the imbalance between MMPs and TIMPs is necessary to regulate the collagen turnover of the cardiac ECM. A disturbance to carefully balanced MMP activity can have significant consequences. Monitoring MMP and TIMP profiles may offer a bedside cutting-edge method for keeping track of the myocardial remodelling process following surgical or percutaneous treatments. It might serve as a reliable and unbiased biomarker by the patient’s bedside. Additionally, therapeutic drugs that target MMPs/TIMPs and fibrosis in HCM patients at presentation are still needed.

## 5. Conclusions

Understanding the molecular processes that control the cardiac ECM’s collagen turnover allows for a non-invasive evaluation of HCM patients through the monitoring of MMPs and TIMPs. Moreover, in order to scale up any potential therapeutic treatment for HCM that targets MMPs, large prospective multicenter trials are required.

## Figures and Tables

**Figure 1 biomolecules-13-00665-f001:**
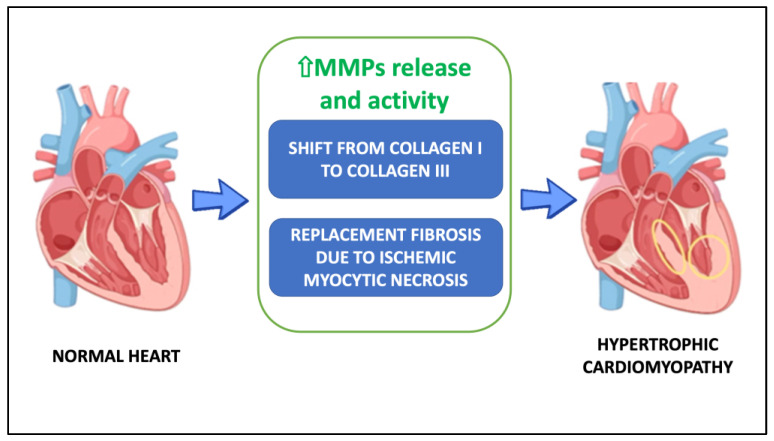
Potential mechanisms underlying hypertrophic cardiomyopathy. Abbreviations: MMPs, metalloproteinases.

**Table 1 biomolecules-13-00665-t001:** Characteristics of the included studies.

Study, Year [Ref.]	Study Design	N° of Patients	Population	Control Group	MMP	Other Factos	Outcomes	Main Findings
Bradham, 2002 [10]	Prospective observational NRCT	51	Patient with HOCM submitted to alcohol septal ablation (51 pts)	-	MMP-2, MMP-8, MMP-9, MMP-13, TIMP-1	CK CK-MB1	To analyse the temporal profile of MMPs plasma levels after alcohol-induced MI; To evaluate the relationship between alcohol-induced MI and MMPs plasma levels.	MMP-2 increased at 4 h post alcohol injection. MMP-8 increased by 6 h post injection and remained elevated for up to 60 h post injection. MMP-9 increased at 6 h and remained elevated for up to 50 h post injection. MMP-13 decreased at 24 h post injection. TIMP-1 increased at late time points post alcohol injection but without statistical significance. MMP-8/TIMP-1 and MMP-9/TIMP-1 ratios increased post alcohol septal ablation.
Lombardi, 2003 [11]	Prospective observational NRCT	50	Patient with HOCM (36 pts)	Healthy subjects (14 pts)	MMP-1, MMP-2, MMP-9, TIMP-1	ICTP, PICP, PINP	To evaluate myocardial collagen turnover; To assess myocardial collagen turnover on diastolic function.	HOCM patients had higher MMPs plasma levels than the control group. No correlation between biomarkers of collagen turnover and severity of left ventricular hypertrophy.
Fassbach, 2004 [12]	Prospective observational NRCT	64	Patient with HOCM (26 pts)	Healthy subjects (38 pts)	MMP-1, TIMP-1	ICTP, PICP	To evaluate myocardial collagen turnover; To assess myocardial collagen turnover on diastolic function.	TIMP-1 was elevated in HOCM patients. MMP-1 was higher in HOCM patients. MMP-1/TIMP-1 ratio was lower in HOCM patients.
Noji, 2004 [13]	Prospective observational NRCT	78	Patient with HOCM (28 pts)	Healthy subjects (50 pts)	MMP-2, MMP-3, MMP-9, TIMP-1, TIMP-2	-	To test if MMPs are altered in patients with HOCM and are associated with LV remodelling.	MMP-2 and TIMP-2 were higher in HOCM patients with EF < 25% than those with EF ≥ 25% and the control group. TIMP-1 was higher in the HOCM patients than the control group. MMP-3 and MMP-9 were similar among groups.
Stroud, 2005 [14]	Prospective observational NRCT	34	Patient with HOCM submitted to alcohol septal ablation (18 pts)	Healthy subjects (16 pts)	TIMP-4	-	To assess if variation in plasma TIMP-4 levels occurs after alcohol-induced myocardial infarction.	TIMP-4 increased in HOCM patients. TIMP-4 increased at 10 h post alcohol injection and decreased after 30 h.
Roldàn, 2008 [15]	Prospective observational NRCT	125	Patient with HOCM (67 pts)	Healthy subjects (58 pts)	MMP-1, MMP-2, MMP-9, TIMP-1	NT-proBNP	To evaluate the correlation between MMPs and myocardial collagen turnover, fibrosis, functional status, and prognosis.	MMP-2 and MMP-9 were higher in HOCM patients. MMP-2 was associated with dyspnoea and correlated with MMP-9 and NT-proBNP. MMP-9 was associated with fibrosis.
Saura, 2009 [16]	Prospective observational NRCT	73	Patient with HOCM (73 pts)	-	MMP-2, TIMP-1	NT-pro-BNP, C-reactive protein	To assess the influence of left atrial volume on exercise performance to MMP2 and TIMP-1.	Enlarged left atrial volume is associated with MMP-2. Left atrial volume is significantly associated with TIMP-1.
Kitaoka, 2010 [17]	Retrospective observational NRCT	41	Patient with HOCM (41 pts)	-	MMP-2, MMP-9, TIMP-1	BNP	To evaluate the role of MMPs levels in the LV remodelling process; To assess the MMPs as a predictor of heart failure in HOCM patients.	MMP-2 levels were higher in HOCM patients with severe symptoms; MMP-2 and TIMP-1 levels were positively associated with LV end-systolic and left atrial dimensions, and inversely related to LV EF. MMP-2 levels were positively related to BNP levels; MMP-9 were not related to echocardiographic parameters and plasma BNP levels.
Kitaoka, 2011 [18]	Prospective observational NRCT	16 *	Patient with HOCM (16 pts)	-	MMP-2, MMP-9, TIMP-1	BNP	To evaluate whether MMPs plasma levels were related to LV remodelling; To observe whether MMPs levels were related to the deterioration in left ventricular systolic function in HOCM patients with preserved LV EF.	MMP-2 levels were inversely related to LV EF; MMP-9 levels were inversely related to LV end-diastolic dimension and positively related to the maximum LV wall thickness.
Zachariah, 2012 [19]	Prospective observational NRCT	45	Patient with HOCM (45 pts)	-	MMP-1, MMP-2, MMP-3, MMP-9, TIMP-1, TIMP-2, TIMP-3, TIMP-4	-	To evaluate the association between ventricular arrythmia and MMPs.	Patients with ventricular arrythmia had significantly higher MMP-3 levels than patients without arrythmia. Ventricular arrythmia was independently associated with MMP-3.
Fucikova, 2016 [20]	Prospective observational NRCT	30	Patient with HOCM (17 pts)	Healthy subjects (17 pts)	MMP-2, MMP-9, TIMP-1, TIMP-2	BNP, fibronectin	To assess the abundance of proteins and plasma fibronectin in HOCM patients.	TIMP-2 levels were higher in HOCM patients; MMP2, MMP-9, and TIMP-1 were similar between groups.
Munch, 2016 [21]	Retrospective observational NRCT	54	Patient with HOCM (54 pts)	-	MMP-1, MMP-2, MMP-3, MMP-9, TIMP-1	-	To evaluate if myocardial collagen turnover changes according to gender; To evaluate myocardial collagen turnover proteins and cardiac events.	MMP-9 was associated with fibrosis and cardiac events in females; Increased MMP-2 levels in females were associated with lower fibrosis; MMP-3 levels were positively associated with cardiac events; No association was detected for MMP-1 and TIMP-1.
Fernlund, 2017 [22]	Prospective observational NRCT	105	Patient with HOCM (39 pts)	Healthy subjects (66 pts)	MMP-9	-	To evaluate whether serum biomarkers reflect myocardial remodelling in the early stage of HOCM.	MMP-9 levels were increased in the HOCM patients and correlated to the LV mass index.
Yang, 2019 [23]	Prospective observational NRCT	52	Patient with HOCM submitted to surgical myectomy (52 pts)	-	MMP-2, MMP-9, TIMP-1	-	To assess the biomarkers of myocardial fibrosis in HOCM patients submitted to cardiac surgery.	TIMP-1 plasma levels were correlated to myocardial content; MMP-2 and MMP-9 were not significantly correlated with those of myocardial tissues; Plasma MMP-9 levels were found to be significantly related to TIMP-1 levels; Plasma MMP-2 levels were positively correlated with collagen volume fraction.
Qian, 2020 [24]	Prospective observational NRCT	30	Patient with HOCM submitted to percutaneous intramyocardial septal radiofrequency ablation (30 pts)	-	MMP-2	-	To assess myocardial fibrosis after percutaneous intramyocardial septal radiofrequency ablation in HOCM patients.	At 1-month follow-up, MMP-2 levels were lower than before the procedure.
Bi, 2021 [25]	Prospective observational NRCT	55	Patient with HOCM submitted to surgical myectomy (55 pts)	-	MMP-2, MMP-9, TIMP-1	-	To demonstrate the relationship among myocardial collagen turnover proteins, microvascular remodelling, and myocardial fibrosis in HOCM patients. To assess the prognostic value of collagen-related biomarkers in HOCM patients after cardiac surgery.	Plasma MMP-2 correlated with left atrium diameters; Septal thickness was associated with the plasma MMP-9/TIMP-1 ratio, MMP-2/TIMP-1 ratio, and TIMP-1 levels; LVOT pressure gradient was correlated with MMP-2/TIMP-1 ratio; Microvascular density was inversely correlated with MMP-2/TIMP-1 ratio; Microvascular density was higher in patients with lower MMP-2 levels; MMP-9 and MMP-9/TIMP-1 were not related to microvascular density; No significant correlations were found between myocardial MMPs and plasma MMPs.

* Among 16 patients, 11 patients were included in a previous study by the same group (Kitaoka 2010). Abbreviations: CK, creatine kinase; EF, ejection fraction; HOCM, hypertrophic obstructive cardiomyopathy; ICTP, collagen I carboxy-terminal telopeptide; LV, left ventricle; MI, myocardial infarction; MMP, metalloproteinase; NRCT, not randomized controlled trial; NT-proBNP, N-terminal pro B-type natriuretic peptide; PICP, procollagen type I carboxy-terminal propeptide; PINP, procollagen type I amino-terminal propeptide; RCT, randomized controlled trial; TIMP, tissue inhibitors of metalloproteinases.

**Table 2 biomolecules-13-00665-t002:** Matrix metalloproteinases and tissue inhibitor of matrix metalloproteinases identified in patients with hypertrophic cardiomyopathy.

Subgroup	MMP	Nomenclature	Mass (kDa)	Substrate	References
Interstitial collagenase	MMP-1	Fibroblast collagenase	52	Collagens I, II, III, VI, VIII and X, gelatin, aggrecan, MMP-2, MMP-9	[11,12,15,19,21]
	MMP-8	Neutrophil collagenase or Collagenase 2	75	Collagens I, II, III, V, VII, VIII and X, gelatin, aggrecan	[10]
	MMP-13	Collagenase 3	54	Collagens I, II, III and IV, gelatin, aggrecan	[10]
Gelatinases	MMP-2	Gelatinase A	72	Gelatin, collagen types I, IV, V, VII, X, XI and XIV, elastin, fibronectin, aggrecan	[10,11,13,15,16,17,18,19,20,21,23,24,25]
	MMP-9	Gelatinase B	92	Gelatin, collagen types IV, V, VII, and X, elastin	[10,11,13,15,17,18,19,20,21,22,23,25]
Stromelysins	MMP-3	Stromelysin 1	57	Collagens III, IV, IX and X, gelatin, aggrecan, MMP-1, MMP-7, MMP-8, MMP-9 and MMP-13, laminin, fibronectin, non-helical collagen	[13,19,21]
Glycoproteins	TIMP-1	-	28	all MMPs except MMP-14	[10,11,12,13,15,16,17,18,19,20,21,23,25]
	TIMP-2	-	21	all MMPs	[13,19,20]
	TIMP-3	-	24	all MMPs	[19]
	TIMP-4	-	23	all MMPs	[14,19]

Abbreviations: MMP, matrix metalloproteinases; TIMP, tissue inhibitor of matrix metalloproteinase.

## Data Availability

PubMed: https://pubmed.ncbi.nlm.nih.gov/ (accessed on 1 November 2022); Scopus: https://www.scopus.com/home.uri (accessed on 1 November 2022).

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
