# Peer review of "Metalloproteinases and Hypertrophic Cardiomyopathy: A Systematic Review"

_biomolecules, 2023, doi:10.3390/biom13040665_

Round 1

Reviewer 1 Report

The review is clear, comprehensive and relevance to the field, and the author summarize the data and provide a possibility that MMPs play a role of the HCM onset or are markers of HCM.

The authors are trying to summarize and discuss the existing knowledge of MMPs profile in patients with HCM from previous published clinical data. The authors found that MMPs are elevated in HCM patients in multiple studies. This review is the first to offer a thorough explanation of the role of metalloproteinases in HCM patients.

There are some room to improve, such as discussions of heart specific ECMs, MMPs and TIMPs are insufficient.  Some of the terms or claims are inaccurate; “Histologically, HCM is determined by an altered collagen turnover of the extracellular matrix”; “The most common cause of this condition in patients is left ventricular outflow obstruction”.

The sentence “Hypertrophic cardiomyopathy (HCM) is a genetic condition characterized by obstruction of the left ventricular outflow” is not accurate, as only one third of the HCM shows obstruction of the left ventricular outflow.

HCM is defined based on its phenotypic features: cardiac hypertrophy, a nondilated left ventricle, and a preserved or an enhanced left ventricular ejection fraction, but the authors trying to say “MMP-2 was found increased in HCM patients with low ejection fraction and was associated with adverse cardiac chambers remodelling”.

Table 1&2 can be formatted better.

The discussion should be split to several topics with a subtitle for each topic.

Reviewer 2 Report

It seems to me that there is a technical problem with the presentation of the results of the literature review in this article (in table 1 the results of only one study are presented, and in table 2 the right edge is cut off and it is not clear which data are missing). I ask the authors to confirm the correctness of the tables.

Reviewer 3 Report

This systematic review is interesting. I understand the importance of this REVIEW, but there are some issues with this manuscript.

1) Authors menthoned that ”The systematic review was performed according to PRISMA (Preferred Reporting Items for Systematic Reviews and Meta- Analyses) guidelines.  ” in the present review. However, I think that this review is not so much this guideline. 

2) A more detailed description of the literature selection is also required.

3) Are you following the rules for posting? It is necessary to check the description method.

4) I think the limitations of the study should be described in more detail.

Round 2

Reviewer 2 Report

The authors presented a systematic review of the role of metalloproteinases as diagnostic and prognostic biomarkers of HCM. The methodological part of the review is described well, the authors also added information to the tables, according to my review No. 1. At the same time, the relevance of this topic and the interest of readers in the work is not great.

Reviewer 3 Report

There is no more comments for this.